

# Gastrointestinal transit time and heart rate variability in patients with mild acquired brain injury

Johannes Enevoldsen[1], Simon T. Vistisen[2], Klaus Krogh[3], Jørgen F. Nielsen[4], Karoline Knudsen[5], Per Borghammer[5] and Henning Andersen[1]

[1] Department of Neurology, Aarhus University Hospital, Aarhus, Denmark
[2] Research Centre for Emergency Medicine, Aarhus University, Aarhus, Denmark
[3] Neurogastroenterology Unit, Department of Hepatology and Gastroenterology, Aarhus Univeristy Hospital, Aarhus, Denmark
[4] Hammel Neurocenter and University Clinic, Aarhus University, Aarhus, Denmark
[5] Department of Nuclear Medicine & PET-Centre, Aarhus University Hospital, Aarhus, Denmark

Corresponding author
Johannes Enevoldsen, johene@rm.dk, johannes.ne@gmail.com

## ABSTRACT

**Background**. Constipation is suspected to occur frequently after acquired brain injury (ABI). In patients with ABI, heart rate variability (HRV) is reduced suggesting autonomic dysfunction. Autonomic dysfunction may be associated with prolonged gastrointestinal transit time (GITT). The primary aim of this study was to investigate if GITT is prolonged in patients with ABI. Secondarily, HRV and its correlation with GITT was investigated.

**Methods**. We included 25 patients with ABI (18 men, median age: 61.3 years, range [30.7–74.5]). GITT was assessed using radio-opaque markers and HRV was calculated from 24-hour electrocardiograms. Medical records were reviewed for important covariates, including primary diagnosis, time since injury, functional independence measure, and use of medication. The GITT assessed in patients was compared to a control group of 25 healthy subjects (18 men, median age: 61.5 years, range [34.0–70.9]).

**Results**. In ABI patients, the mean GITT was significantly longer than in healthy controls (2.68 days, 95% CI [2.16–3.19] versus (1.92 days, 95% CI [1.62–2.22], $p = 0.011$)). No correlation was found between HRV and GITT.

**Conclusion**. Patients with mild to moderate ABI have prolonged GITT unrelated to the HRV.

## INTRODUCTION

Acquired brain injury (ABI) is a frequent cause of mortality and disability. In 2009, the incidence of hospitalizations due to ABI was approximately 400 per 100,000 in Denmark (*National Board of Health, 2011*), and in 2012, the mean European incidence of hospitalizations due to traumatic brain injury (TBI) was 287.2 per 100,000 (*Majdan et al., 2016*).

Constipation and prolonged gastrointestinal transit time (GITT) are very common complications to acute and chronic spinal cord lesions (*Han, Kim & Kwon, 1998*; *Krogh, Mosdal & Laurberg, 2000*), but little is known about bowel function after ABI.

Two studies including Korean patients with ABI or stroke found that GITT was associated to subjective symptoms of constipation (*Lim et al., 2012*; *Yi et al., 2011*). However, none of the studies included a healthy control group. Other data indicate that gastric emptying is delayed in patients with TBI (*Kao et al., 1998*). In animal studies, contractile activity was reduced and transit time of the small intestine was prolonged after TBI (*Olsen et al., 2013*; *Wang, Liu & Yang, 2011*). A study, comparing patients with stroke to patients with orthopedic injuries and similar immobility, showed that patients with stroke had a higher risk of *de novo* constipation (assessed by the Rome II questionnaire) (*Bracci et al., 2007*). This suggests that ABI causes bowel dysfunction. It is likely that impaired gastrointestinal motility may be caused, at least in part, by injuries to the central autonomic nervous system. Other causes of gastrointestinal dysmotility after ABI may include immobility and side effects of medication.

Autonomic function can be assessed, non-invasively, by measuring heart rate variability (HRV). High frequency variability in the heart rate is mediated primarily by changes in the parasympathetic (vagal) tone on the heart (*Malik et al., 1996*; *Metelka, 2014*). The parasympathetic nervous system modulates bowel motility also (*Furness, 2012*), and as recent studies reported that HRV is lowered in patients with ABI (*Keren et al., 2005*; *King et al., 1997*; *Vistisen et al., 2014*), it is relevant to investigate the correlation between HRV and GITT in these patients.

To enable identification of residual gastrointestinal symptoms or impaired motility, despite presumed optimal laxative treatment, the aim of this study was to compare GITT in patients with ABI and healthy control subjects. Furthermore, we aimed to investigate whether there is an association between GITT and HRV in patients with ABI.

## METHOD

### Subjects

This study was approved by the regional research ethics committee (application no: 49007) and by the Data Protection Agency (jr.no: 2007-58-0010). The study was registered at ClinicalTrials.gov (identifier: NCT02428790).

Twenty-five patients undergoing rehabilitation for ABI were included from Hammel Neurocenter and University Clinic between June and November 2015.

Inclusion criteria were: Age 18 to 79 years, ABI within one year before inclusion (i.e., stroke including subarachnoid haemorrhage, anoxic brain injury or moderate to severe TBI (Glasgow Coma Scale (GCS) score 3–12 at admission to the intensive care unit)), ability to swallow the capsules used to assess GITT and sufficient cognitive performance to give informed consent.

Exclusion criteria were: Major abdominal disorders, severe acute comorbidity, cancer, pregnancy, autonomic neuropathy, or other known neurological disorder.

Informed consent was obtained.

Control subjects were recruited from two historic datasets from our unit including 38 subjects. One set consisted of healthy hospital and university employees (a subset of these have previously been published (*Krogh, Mosdal & Laurberg, 2000*)), and the other consisted of an elderly control group, matched to a study population of patients with Parkinson's disease (*Knudsen et al., 2017*). No control subject used prescription laxatives or had gastrointestinal diseases. Control subjects were selected blindly from the two datasets to match the age and sex of the patients with ABI on a group level, using the following method: From the total set of control subjects, subjects older than 79 years were excluded. The set was split in males and females and ordered by age. With a ratio of 2.6 males per female (as in the patient cohort), the oldest control subjects were selected until the median age matched the patient cohort.

### Gastrointestinal transit time

Total GITT was assessed with radio-opaque markers as described by (*Abrahamsson, Antov & Bosaeus, 1988*). The measurement is a 7 day procedure, where the patients, on each of days 1–6, ingest one capsule containing ten radio-opaque markers (Colon Transit, REF:CTT6V10). An X-ray examination of the abdomen is performed on day 7. Ingestion of capsules and the X-ray examination was done at noon, except for one patient, who, for convenience, had both done at 8 am.

GITT was estimated using the published equation: $GITT = (M + 0.5*D)/D$, in which GITT is the gastrointestinal transit time in days, $M$ is the number of markers counted on the X-ray and $D$ is the daily dose of markers ($D = 10$).

In other publications, GITT is often denoted as colonic transit time (CTT). However, the method assesses the transit time of the entire gastrointestinal system, with colonic transit time accounting for the majority (*Degen, Phillips & Izzo, 1996*).

### Neurogenic bowel dysfunction score

After the end of the inclusion period, one of the authors (JE) contacted the participants, and performed a questionnaire regarding their bowel function, using the Neurogenic Bowel Dysfunction (NBD) score (*Krogh et al., 2006*). Patients who were still admitted were contacted in person, while discharged patients were contacted by phone. In both cases, the interviewer read the questions aloud and registered the patient's answers.

### Heart rate variability

Twenty-four hour electrocardiograms (ECGs) were measured using a wearable 1-lead device (eMotion Faros 90°, Mega Electronics Ltd., Kuopio, Finland), using a sampling rate of 250 Hz. The recordings where performed during the week preceding the X-ray examination.

Heart rate variability measures were calculated from the 24-hour ECG using Kubios HRV (University of Eastern Finland, Kuopio, Finland) (*Tarvainen et al., 2009*). The methods comply with the HRV guideline provided by the Task Force of The European Society of Cardiology and The North American Society of Pacing and Electrophysiology (*Malik et al., 1996*). For each patient, a 24-hour analysis and four 5 min analyses were performed.

The 24-hour analysis included all normal-to-normal intervals (NN). Initial identification of normal beats was performed, using the automatic R wave detection in Kubios HRV. Then manual correction of erroneous or missing R wave markings was performed and ectopic beats were removed. Artefact correction was applied to remove the resulting non-normal intervals. To optimally distinguish between normal and non-normal intervals, individual correction strengths were applied to each ECG recording.

The 5 min analyses were performed approximately at 6 am, 1 pm, 6 pm and 2 am. These time points were chosen because no rehabilitation training was scheduled at these time points, and the patients would usually be physically inactive. Patients were not observed during these 5 min intervals. For each analysis, a 5 min section of the ECG, with little or no non-normal beats and no upwards or downwards trend in RR duration, was selected. Any non-normal beats were removed manually, and the resulting non-normal intervals were removed using the lowest ('very low') artefact correction setting.

Non-parametric calculations (i.e., fast Fourier transforms (FFT)) were used for frequency domain analyses. Standard ranges for low frequency (LF) and high frequency (HF) parts of the spectrum were used (i.e., LF, 0.04–0.15 Hz; HF, 0.15–0.4 Hz).

The main HRV measures were chosen to be the root mean square of successive NN differences (RMSSD) and HF power. These measures are believed to reflect mainly the parasympathetic (vagal) component of the regulation of the heart rate. Other HRV variables are reported for comparison with other studies.

## Covariates

Covariates, including time of injury, type of injury, medication and Functional Independence Measure (FIM), were acquired from the medical records.

Medication use, either scheduled or *pro re nata*, was registered based on patients' medical records on the day of the X-ray examination. Use of oral laxatives, opioid analgesics, statins, selective serotonin reuptake inhibitors (SSRI), and baclofen were registered.

Functional ability of the patients was assessed using FIM. FIM is used for routine scoring at our rehabilitation hospital, and reflects the level of independence in everyday tasks (*Uniform Data System for Medical Rehabilitation, 2012*). Scoring is performed approximately once per month by the team of treating doctors, nurses, physiotherapists and occupational therapists. In this study, we used the FIM scores performed in closest temporal proximity to the X-ray examination.

## Statistical analyses

Results are presented as mean [95% CI], unless otherwise indicated. The difference in GITT and age between patients and healthy control subjects was tested using a Welch $t$-test due to unequal variance. Differences in SD were tested using $F$-tests, and proportions using Pearson's $\chi^2$ test. Linear regression modeling GITT as a function of group (patient vs control), age, and sex was performed.

Correlations including time since injury (at the day of the X-ray examination) and FIM were calculated using Spearman's rank correlation ($\rho$), while correlations between GITT, age and HRV measures were calculated using Pearson product-moment correlation coefficient ($r$).

Comparison of GITT in users with non-users of medication (laxatives, opioids, statins and SSRIs) was done using confidence intervals based on the $t$-distribution. Between users and non-users of laxatives, differences in FIM, time since injury and age were tested using the Wilcoxon-Mann–Whitney test.

Heart rate variability measures for the four 5 min HRV analyses were summarised for each patient (the four analyses are one for each time point). For HRV measures (not including heart rate (HR)) logarithmic transformation was performed prior to calculation of summary statistics, correlations and regressions. Therefore, the reported means are geometric for all HRV measures and arithmetic for HR.

Our power calculation was based on the standard deviation (SD) of GITT among healthy subjects. We assumed that SD of GITT among the patients with ABI would be identical. The target difference in GITT was 1 day, and with a significance level ($\alpha$) of 0.05, 11 patients and 11 control subject would be necessary to reach a statistical power of 0.90.

To ensure sufficient statistical power despite dropouts or higher SD among patients, and to make statistical corrections for age and sex possible, we aimed to include 30 patients with ABI.

Statistical analyses and calculations apart from HRV analysis, was performed using R 3.4.3 (*R Core Team, 2017*) with additional packages (*Bengtsson, 2015*; *Fischer & Pau, 2015*; *Grolemund & Wickham, 2011*; *Hlavac, 2015*; *Robinson, 2015*; *Wickham, 2007*; *Wickham, 2009*; *Wickham & Francois, 2016*).

# RESULTS

During the inclusion period, 103 patients were screened for inclusion. Of these, 45 were eligible for inclusion. Twenty-six patients were included in the study, of which one was later excluded due to an erroneous ABI diagnosis (Fig. 1).

The mean GITT of the 25 patients with ABI was 2.68 [2.17–3.19] days. The 25 healthy control subjects had a lower GITT of 1.92 [1.62–2.22] days ($p = 0.011$). Also, the patients had a higher SD of GITT than the control subjects (1.24 and 0.73 days, $p = 0.011$) (Table 1).

Linear regression of GITT by group (patient vs control), age and sex showed that ABI and higher age were associated with longer GITT, while sex was not a predictor (Table 2 and Fig. 2).

The median time from X-ray examination to FIM scoring was −1 day, IQR [−13–6]. Patients with ABI had a median FIM score of 111, IQR [97–116]. FIM score did not correlate with age ($\rho = -0.21$, $p = 0.31$) or GITT ($\rho = -0.32$, $p = 0.12$).

The median time since injury was 71 days, IQR [39–131]. Time since injury did not correlate with GITT ($\rho = -0.06$, $p = 0.77$) or FIM ($\rho = 0.02$, $p = 0.91$).

Twelve (48%) patients with ABI received oral laxatives. GITT for these patients were not significantly longer than GITT for patients who did not use oral laxatives (+0.33 [−0.71–1.37] days). There was no significant difference in age (−1.1 years, $p = 0.77$) or time since injury (−29.6 days, $p = 0.22$) between the two groups (laxative users–non-users). Laxative users did have a significantly lower FIM score (difference: 19.7, $p = 0.02$), which remained significant after subtraction of the bowel function subscore from the total

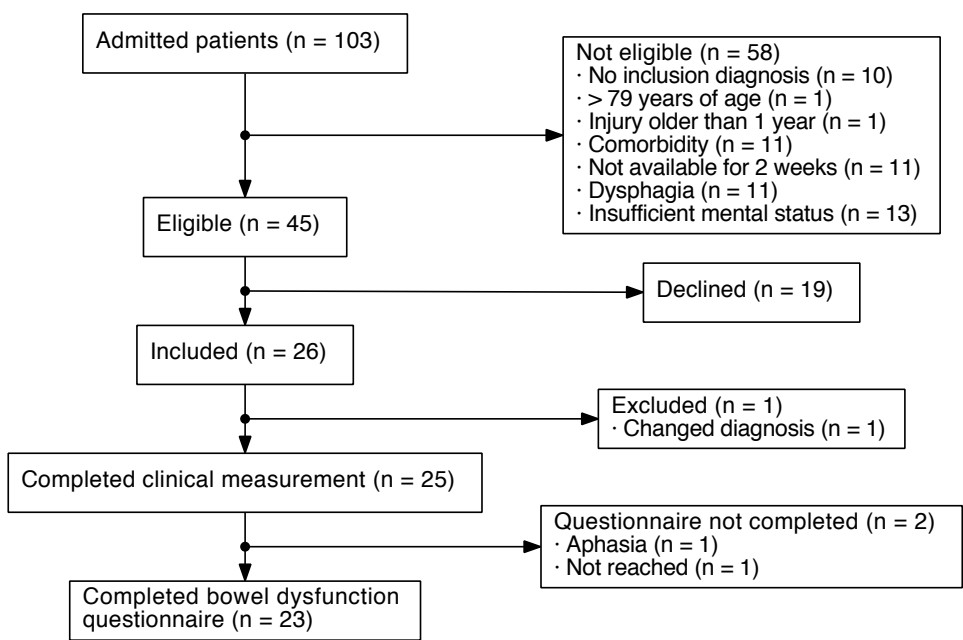

**Figure 1  Flowchart of patients with acquired brain injury (ABI) included in the present study.**

FIM score (difference: 18.6, $p = 0.02$). Of the 13 patients with ABI who did not have a prescription for oral laxatives, 10 (77%) received an oral laxative at some point between their injury and the X-ray examination.

Opioid analgesics, statins and SSRIs were used by 4 (16%), 11 (44%) and 8 (32%) of the patients respectively. None of the patients used baclofen. Neither opioid, statin nor SSRI users had a longer GITT than patients who did not use these medications (opioid users, −0.42 [−1.84–0.99] days; statin users, 0.60 [−0.42–1.63] days; SSRI users, 0.36 [−0.75–1.47] days).

### Neurogenic bowel dysfunction score

The neurogenic bowel dysfunction score (NBD) was completed by 23 patients with ABI. The median time from injury to completion of the questionnaire was 163 days, range [38; 395] and the median time from the X-ray examination to completion of the questionnaire was 82 days, range [8–176].

One patient with ABI had a 'moderate' bowel dysfunction (NBD score = 10) and reported that her gastrointestinal function had 'some' impact on her quality of life. The remaining 22 patients had either no or 'very minor' bowel dysfunction (mean NBD score = 0.5, range [0; 5]) and reported that this had 'no' or 'little' impact on quality of life.

### Heart rate variability

The geometric mean HF power of the 5 min analyses of ECGs was 90.7 [50.5–162.6] ms$^2$. The geometric mean RMSSD was 18.1 [13.4–24.6] ms. No significant correlation was found between GITT and the logarithm of HF power ($r = −0.117$ [−0.490–0.292]) or

**Table 1  Subject characteristics and results.**

| | Patients with acquired brain injury ($n=25$) | Controls ($n=25$) | $p$ |
|---|---|---|---|
| Sex, males | 18 (72%) | 18 (72%) | 1 |
| Age (Years) | | | |
|    Median (IQR) | 61.3 (51.7–68.2) | 61.5 (54.0–65.1) | 0.74[a] |
| Type of injury | | | |
|    Ischemic stroke | 12 (48%) | – | |
|    Hemoragic stroke | 2 (8%) | – | |
|    Subarachnoid hemorrhage | 7 (28%) | – | |
|    Traumatic brain injury | 3 (12%) | – | |
|    Anoxic brain injury | 1 (4%) | – | |
| Days since injury | | | |
|    Median (IQR) | 71 (39–131) | – | |
| FIM score | | | |
|    Median (IQR) | 111 (97–116) | – | |
| Laxative use | 12 (48%) | 0 (0%) | |
|    Magnesium oxide | 10 (40%) | 0 (0%) | |
|    Macrogol | 2 (8%) | 0 (0%) | |
|    Sodium picosulfate | 2 (8%) | 0 (0%) | |
|    Bisacodyl | 1 (4%) | 0 (0%) | |
| Total GITT (days) | | | |
|    Mean (SD) | 2.68 (1.24) | 1.92 (0.73) | 0.011[a] |

**Notes.**
FIM, Functional independence measure; GITT, gastrointestinal transit time.
[a] Welch $t$-test.

**Table 2  Multiple linear regression, modeling gastrointestinal transit time by presence of acquired brain injury (ABI; 25 patients and 25 healthy controls), age and sex.**

| | Coefficient (95% CI) |
|---|---|
| ABI | 0.798[**] (0.249; 1.347) |
| Age (years) | 0.035[*] (0.008; 0.062) |
| Male | −0.233 (−0.844; 0.379) |
| Constant | 0.025 (−1.648; 1.698) |

**Notes.**
[*] $p < 0.05$.
[**] $p < 0.01$.
$R^2 = 0.248$; adjusted $R^2 = 0.199$.

GITT and the logarithm of RMSSD ($r = -0.119$ [$-0.491$–$0.290$]). One patient had several periods with arrhythmia, making only 5 min analyses possible. Further results from the HRV analyses are shown in Tables 3 and 4.

## DISCUSSION

This study aimed to characterise GITT in patients with ABI and compare findings to healthy control subjects. Previous studies in patients with ABI, although using similar

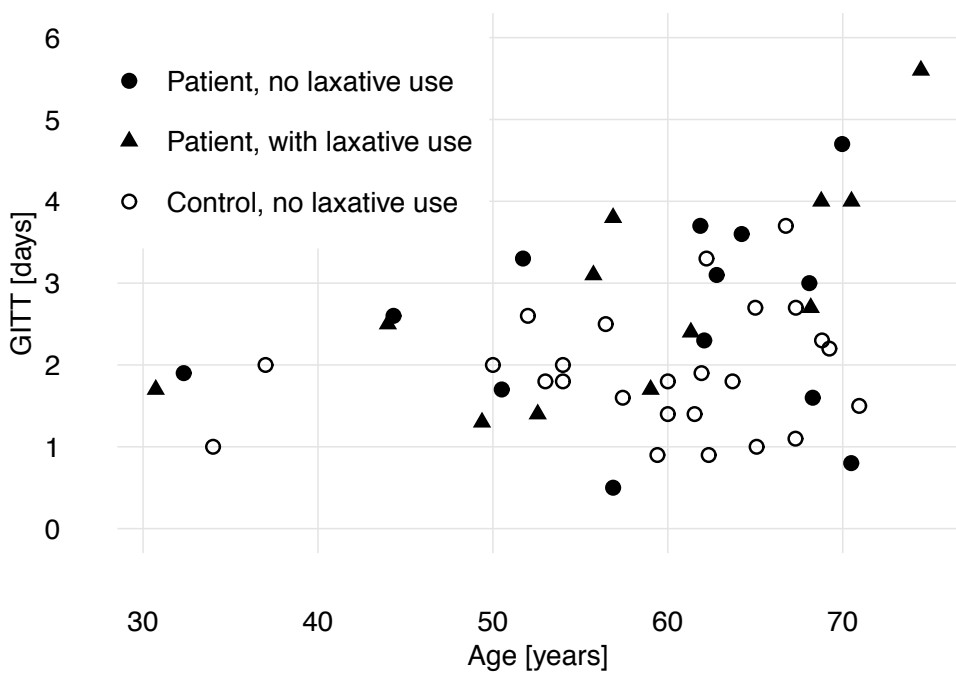

**Figure 2** Gastrointestinal transit time (GITT) and age of patients with acquired brain injury and healthy control subjects.

**Table 3** **Heart rate variability in patients with acquired brain injury.** For each patient, a 24-hour analysis, and a geometric mean of four 5 min analyses measured at 1 pm, 6 pm, 2 am and 6 am was calculated.

| | Geometric mean (95% CI) | |
|---|---|---|
| | 24-hour analysis ($n = 24$) | 5 min analysis ($n = 25$) |
| Heart rate[a] (min$^{-1}$) | 72.3 (67.4; 77.1) | 67.1 (61.9; 72.3) |
| SDNN (ms) | 126.7 (106.5; 150.8) | 29.3 (23.7; 36.2) |
| RMSSD (ms) | 20.5 (15.2; 27.6) | 18.1 (13.4; 24.6) |
| LF power (ms$^2$) | 373.2 (217.4; 640.6) | 217.0 (131.4; 358.3) |
| HF power (ms$^2$) | 126.9 (72.5; 222.1) | 90.7 (50.5; 162.6) |
| Total power (ms$^2$) | 13,503 (8,792; 20,739) | 782 (518; 1,183) |
| LF/HF | 2.9 (2.2; 3.9) | 2.4 (1.7; 3.4) |

Notes.
[a] Arithmetic mean.

SDNN, standard deviation of normal-to-normal intervals; RMSSD, root mean square of successive differences; LF, low frequency; HF, high frequency.

methodology, did not perform a direct comparison to healthy control subjects (*Lim et al., 2012*).

Patients with ABI had significantly longer GITT than healthy control subjects, but with substantial overlap. While the mean GITT of 2.68 days is significantly longer than normal, it is only modestly increase compared with the mean GITT of >4 days found in patients with acute spinal cord lesions (*Krogh, Mosdal & Laurberg, 2000*). The NBD score revealed limited clinical bowel dysfunction in the patient group. This is surprising compared to a
**Table 4   Correlations between the mean of four 5 min heart rate variability analyses and gastrointestinal transit time (GITT), age, and functional independence measure (FIM).**

|  | GITT (days)[a] | Age (years)[a] | FIM[b] |
|---|---|---|---|
| HR | −0.027 | −0.145 | −0.388 |
| log(SDNN) | −0.247 | −0.504[*] | 0.370 |
| log(RMSSD) | −0.119 | −0.309 | 0.372 |
| log(HF power) | −0.117 | −0.330 | 0.375 |
| log(LF power) | −0.294 | −0.528[**] | 0.350 |
| log(LF/HF) | −0.232 | −0.212 | −0.059 |

Notes.

[*]$p < 0.05$.

[**]$p < 0.01$.

[a]Pearson's product-moment correlation ($r$).

[b]Spearman's rank correlation ($\rho$).

SDNN, standard deviation of normal-to-normal intervals; RMSSD, root mean square of successive differences; LF, low frequency; HF, high frequency.

study of *de novo* constipation on hemiplegic stroke patients, which found constipation in 30% of patients (*Bracci et al., 2007*). Their time from injury to interview was longer (254 days, IQR [138–565] against 163 days, IQR [109–213] in our study). However, only 24.4% of patients used laxatives, against 48% in our study, and the questionnaire used to assess bowel function (Rome II) was different. Advanced age was related to an increased GITT, which is a common, though not entirely consistent, finding in other studies (*Graff, Brinch & Madsen, 2001*).

There are large global differences in GITT, with western populations having a markedly longer mean transit time than Asian (*Jung, Kim & Moon, 2003*). Since alternate and shorter protocols are often used for measuring GITT in populations with faster transit times, direct comparisons among studies are difficult (*Ghoshal, Sengar & Srivastava, 2012*). The healthy control subjects presented in our study had a mean GITT similar to that reported in a Swedish study (*Abrahamsson, Antov & Bosaeus, 1988*). On the other hand, (*Lim et al., 2012*) showed that Korean patients with ABI, who were clinically constipated, had a GITT similar to the healthy control subjects in our study. The mean GITT was 1.94 days in Korean patients with ABI and 1.92 days in our healthy control subjects (*Lim et al., 2012*). Thus, the inclusion of local control subjects seems mandatory in GITT studies.

We found no correlation between HRV and GITT which may be due to low statistical power. The use of laxatives may have hidden any correlation by narrowing the variability in GITT. It is also possible that the correlation does not exist. The HRV variables, HF and RMSSD, predominantly reflect parasympathetic (vagal) tone on the sinoatrial node. The enteric nervous system, which controls gastrointestinal motility, responds not only to parasympathetic input but also to local and regional stimulation such as mechanical stretch of the intestinal wall and endocrine hormones. Such regional and local stimulation may be more pronounced determinants of gastrointestinal motility than parasympathetic tone, thereby masking any correlations with HRV. It would be important to investigate the correlation between GITT and HRV in more severely injured patients with ABI.
In a previous study from our rehabilitation hospital we compared HRV in 49 patients with ABI with 49 healthy control subjects (*Vistisen et al., 2014*). The patients with ABI were included from the semi-intensive care rehabilitation unit. They were all bedridden and had a markedly lower FIM score. The mean HRV of the patients in the present study lie between the patients and the controls from the previous study, which suggests a less severe autonomic dysfunction in the present patients as compared to more severely injured patients with ABI (HF power: Healthy controls, 290 [211–399] ms$^2$; patients in the current study, 90.7 [50.5–162.6] ms$^2$; patients in semi-intensive care rehabilitation, 8.9 [5.4–14.8] ms$^2$).

Some limitations need to be addressed.

First, for ethical reasons, we accepted laxative use during the GITT measurement, among the patients with ABI. This may have resulted in an underestimation of the difference in GITT between patients and healthy control subjects.

Second, we did not include patients with dysphagia or mental impairment. Thereby only patients with milder ABI or late in their rehabilitation have been included. Adding to this, patients with more severe illness may be less likely to participate, due to fatigue. Indeed, our rehabilitation centre treats patients who are generally more disabled than the cohort in this study (*Stubbs et al., 2014*). The results are therefore not necessarily representative of patients in the acute phase after ABI and patients with severe disability.

Third, though we did exclude patients with autonomic neuropathy or other neurological diseases apart from ABI, patients were not examined for unrecognised neuropathy. As diabetic patients are at risk of both stroke and autonomic neuropathy, this may have contributed to the prolonged GITT and low HRV observed in these patients. The prevalence of diabetes in our study population is unknown.

Other limitations include the heterogeneity of the patients included, the late application of the bowel dysfunction questionnaire and some characteristics about the HRV measurement discussed below. We have not reported the severity of patients' injuries at presentation, as different severity scores are used for each group of injuries. Also, severity scores were not available for all patients. We believe that the FIM score is a good indicator of the patients' condition at the time of GITT measurement.

Heart rate variability can be confounded by endogenous factors. Most importantly, respiratory HRV components (i.e., HF and RMSSD) are lowered with increasing respiratory rate and heart rate (*Weippert et al., 2015*). To minimise the effect of physical activity on HRV, the 5 min analyses were performed on ECG recordings of periods, where the patients were scheduled to be resting.

This study does not cover whether the bowel dysfunction is a direct effect of the ABI or secondary to the changes in daily living following ABI (e.g., immobility and changes in diet). It may well be a combination and the clinical issue is important to address regardless of the causation.

We consider that prolonged GITT, despite common use of laxatives, support the clinical practice of laxative prescription to patients with mild impairment after ABI. However, the observational nature of this study does not prove the usefulness of laxatives in this patient

group. We hope this new evidence on bowel function patients with ABI will reinforce the importance of managing gastrointestinal impairment in these patients.

## CONCLUSION

Gastrointestinal transit time (GITT) was significantly prolonged in patients with mild impairment after ABI compared with healthy controls. The difference may have been underestimated due to laxative use in half of the patients. No correlation between GITT and HRV was found. Further studies are needed to assess bowel function in the acute phase after ABI.

## ACKNOWLEDGEMENTS

Thanks to Department of Radiology, Regional Hospital Silkeborg, for support and education and to the ward staff at Hammel Neurocenter and University Clinic for invaluable help with the day-to-day operation of the study.

### Funding

The study was supported under Lunbeckfonden (scholarship granted by the Danish Neurological Society), under Grosserer L.F. Foghts Foundation, and under the BEVICA Foundation. The funders had no role in study design, data collection and analysis, decision to publish, or preparation of the manuscript.

### Grant Disclosures

The following grant information was disclosed by the authors:
Lunbeckfonden (scholarship granted by the Danish Neurological Society).
Grosserer L.F. Foghts Foundation.
BEVICA Foundation.

### Competing Interests

The authors declare there are no competing interests.

### Author Contributions

- Johannes Enevoldsen conceived and designed the experiments, performed the experiments, analyzed the data, prepared figures and/or tables, authored or reviewed drafts of the paper, approved the final draft.
- Simon T. Vistisen conceived and designed the experiments, contributed reagents/-materials/analysis tools, authored or reviewed drafts of the paper, approved the final draft.
- Klaus Krogh, Jørgen F. Nielsen and Henning Andersen conceived and designed the experiments, authored or reviewed drafts of the paper, approved the final draft.
- Karoline Knudsen and Per Borghammer contributed reagents/materials/analysis tools, authored or reviewed drafts of the paper, approved the final draft.

## Human Ethics

The following information was supplied relating to ethical approvals (i.e., approving body and any reference numbers):

The Regional Research Ethics Committee of Central Denmark granted Ethical approval to carry out the study within its facilities (application no: 49007).

## Ethics

The following information was supplied relating to ethical approvals (i.e., approving body and any reference numbers):

The Data Protection Agency granted Ethical approval to carry out the study within its facilities (journal no: 2007-58-0010).

## Data Availability

GitHub: https://github.com/JohannesNE/gitt-after-braininjury.

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
