# Peer review of "Gastrointestinal transit time and heart rate variability in patients with mild acquired brain injury"

_PeerJ, doi:10.7717/peerj.4912_

## Round 0.1 · original submission · Major Revisions

Reviewers have came back with their comments and there are major concerns needed to address. Revisions are required before the paper can be considered further. I share similar concerns with other reviewers in that there are a number of confounding factors in your study. Among others include the inclusion of milder type of ABI and use of laxatives. Do consider revising the title of your paper to 'mild ABI'.

Reviewer 1 ·

Basic reporting

This is an interesting convenience sampling study that looked at the gastrointestinal transit time and heart rate variability in TBI patients.

In general, the sentences were good with only minimal grammar errors.

Abstract
- Good abstract with precise sentences.

Intro.
1) The statistics data used fo the ABI incidence was quite old data, nearly a decade. I am sure there is a newer data to support this study.
2) The research gap is well explained.
3) Good discussion and referring to other previous papers but maybe emphasize can also be made on how the study will impact the medical and general community.

Experimental design

Methodology.
1) Good description of inclusion & exclusion criteria
2) However, for TBI, the classification used is quite broad (ie based on GCS at admission to ICU which sometimes is masked by some drugs). Do you have any other classification for usage of TBI classification (ie GCS on first clinic apppointment, Rancho Los Amigos cognitive scale etc?)
3) Any specific classification for Subarachnoid hemorrhage patients included in this study?
4) For exclusion criteria, do you rule out peripheral nerve problem ie pre-existing diabetic neuropathy which is unrelated to the central nervous system?
5) How “blinding” was performed in selecting control subjects?
6) For the neurogenic Bowel dysfunction scoring and FIM scoring, are you using the same person to ask the questionaires? How do you standardised this part of your study?

Validity of the findings

Discussion.
- Good discussion with good explanation on the findings.
- Limitation of the study was explained well.
- Interprets the findings of this study well and good comparison with previous studies in this topic

Additional comments

- This is a good paper, and introduce new finding in the brain injury rehabilitation field.

·

Basic reporting

no comment

Experimental design

no comment

Validity of the findings

no comment

Additional comments

I have reviewed the manuscript “Gastrointestinal transit time and heart rate variability in patients with acquired brain injury”. This is an interesting study evaluating the gastrointestinal transit time after brain injury. I have several queries as listed below.

1. Methods: The authors did not mention whether the patients stopped any laxatives they were on at least 3 days before GITT. However we understand from Results Section that 12 (48%) of patients received laxatives. This is a pitfall.
2. Methods: Please move Neurogenic bowel dysfunction score subtitle after GITT as this will be flowing with the results.
3. Discussion: It may be better to discuss the GITT results first (to the 3rd paragraph), before discussing GITT-correlation.
4. Discussion: Since some patients continued to use laxatives, this probably affected GITT results. May this be one reason for no correlation between HRV and GITT other that low statistical power?
5. Discussion: Authors need to discuss more the correlation with other studies.
6. Discussion: There are several limitations of this study that are already given in the Discussion section. Actually limitations are much widely given than other discussion.
7. Conclusion: Authors need to state her that no correlation was detected between GITT and HRV.

---

## Round 0.2 · accepted · Accept

The reviewers' comments have been adequately addressed

·

Basic reporting

No comment

Experimental design

No comment

Validity of the findings

No comment

Additional comments

The authors have responded my queries made changes on manuscript accordingly.